# Generating Alerts from Breathing Pattern Outliers

**DOI:** 10.3390/s22166306

**Published:** 2022-08-22

**Authors:** Chloé Benmussa, Jessica R. Cauchard, Zohar Yakhini

**Affiliations:** 1School of Computer Science, Reichman University (IDC Herzliya), Herzliya 4610101, Israel; 2Magic Lab, Department of Industrial Engineering and Management, Ben Gurion University of the Negev, P.O. Box 653, Be’er-Sheva 8410501, Israel

**Keywords:** wearables, statistical modelling, outlier detection, alert generation

## Abstract

Analysing human physiological data allows access to the health state and the state of mind of the subject individual. Whenever a person is sick, having a panic attack, happy or scared, physiological signals will be different. In terms of physiological signals, we focus, in this manuscript, on monitoring breathing patterns. The scope can be extended to also address heart rate and other variables. We describe an analysis of breathing rate patterns during activities including resting, walking, running and watching a movie. We model normal breathing behaviours by statistically analysing signals, processed to represent quantities of interest. We consider moving maximum/minimum, the amplitude and the Fourier transform of the respiration signal, working with different window sizes. We then learn a statistical model for the basal behaviour, per individual, and detect outliers. When outliers are detected, a system that incorporates our approach would send a visible signal through a smart garment or through other means. We describe alert generation performance in two datasets—one literature dataset and one collected as a field study for this work. In particular, when learning personal rest distributions for the breathing signals of 14 subjects, we see alerts generated more often when the same individual is running than when they are tested in rest conditions.

## 1. Introduction

Analysing physiological data is becoming more and more important, finding applications in security, safety and social domains. More specifically, accessing a person’s breathing rate facilitates a better understanding of the person’s physiological and mental condition. Abnormalities in the respiration rate can indicate that patients are having trouble maintaining homeostatic control (the body’s internal environment) [1]. Analysing breathing rate can also help detect physiological conditions such as hypoxia, hypercapnia or other respiratory conditions. Another reason why breathing rate is useful is to detect a person’s emotional state. Whenever a person is stressed, happy, frightened or mad, breathing rate is affected. In some cases, verbal communication is limited and being able to know someone’s breathing rate could help communicate a need for assistance. Such a situation occurs in the context of children or of individuals that have verbal communication difficulties. In such cases it would be desirable for physiological signals and, especially, alerts, to be communicated to the outside world through physical wearable materials. Wearable devices that implement this approach and support display behaviours that are responsive to the physiological state of the individual that wears them will potentially become extremely useful in the context of challenging environments, such as scuba-diving, and/or for users as above. Figure 1 and Figure 2 depict the desired behaviour of a potential device that uses an alert generation mechanism to change its colour. The implementation of such a device is the topic of future work.

In this work we address alert generation from respiration rate/pattern data. In a recent work, Lehmann et al. [2] studied properties of heart rate signals and their per person characteristics.

Our project uses statistical modelling to represent the distribution of signals, under a normal behaviour, for any given individual. We emphasise that the distributions for all parameters inferred in this paper are personal. Alerts are generated on a personal basis, based on those inferred distributions. Having learnt a distribution for the user, we generate an alert when the signal captured significantly deviates from the mean (a parameter). In this project, we develop a methodology for monitoring the physiological status of a user.

Our contribution is as follows:An alert generation method, based on statistical modelling;A proof of concept of the alert generation system;Example analysis of two datasets, one literature dataset and one collected for this study.

This paper is organized as follows. In Section 1.1 we discuss related work. In Section 2.1 and Section 2.2 we describe the design of our field study and the other datasets used. In Section 2.3, we detail the algorithmic steps. Finally, we describe our results in Section 3, both from the field study and from the literature data. In Section 4 and Section 5 we discuss open issues and future work.

### 1.1. Related Work

Measuring physiological indicators has been developed on many levels and through many technologies. Communicating physiological indicators through wearables and using machine learning to analyse physiological signals has also been studied and used in practice. In this section we will review work in these domains that is relevant to our current work.

#### 1.1.1. Measuring Breathing Rate

Breathing can be measured using airflow, with a sensor put next to the nose or the mouth, or by detecting chest movement, with sensors placed on the body [3]. Breathing has been extracted using stretch sensors, an inertial measurement unit (IMU) strapped to the chest [4], a head-mounted IMU + egocentric camera [5], and a wrist-mounted IMU [6,7]. Interestingly, non-contact breathing rate measurement can also be accomplished. In [8], for example, the authors use a VICON motion capture system with 16 distinct locations on the trunk to get a high density trunk surface motion. Another approach used a smartphone microphone [9], and [10] investigates the analysis of high-frequency wireless signals. Ramos-Garcia et al. [11] integrated a textile-based stretch sensor with a coverstitch formation into a commercial shirt to monitor breathing patterns. The subjects were asked to do eight different activities and the authors used FFT (Fast Fourier Transform) to analyse the frequency content of the signals. In [12], the authors measured the breathing without contact using the MS Kinect depth sensor, by following the chest movements. They defined a classifier that uses amplitude and dominant frequency to determine whether a person is awake or asleep.

#### 1.1.2. Measuring Heart Rate

There are different ways to measure the heart rate of a person. One approach is using a pulse oximetry sensor. This approach also allows to detect the saturation of peripheral oxygen. The sensor is often positioned on the subject’s fingers using finger-clips [13,14,15,16]. Another way is to put the sensor at the back of a wrist-worn device, as presented by Anliker et al. [17]. Heart rate can also be calculated by analysing electrocardiography (ECG) signals captured with ECG electrodes [18,19]. ECG is also used by [2] to associate users with their characteristic patterns (see more in Section 1.1.4). Continuous-wave Doppler ultrasound devices are also useful is this context. Garverick et al. [20] used this technique combined with wireless transmission to measure the heart rate of a foetus. Finally, some measurement approaches use the micro-vibrations caused by the blood ejection of each heartbeat called ballistocardiography (BCG) or seismocardiography (SCG) [21,22,23]. Bieber et al. [24] integrated an accelerometer on a smartwatch that has to be positioned close to the heart. This allowed them to receive the micro-vibration signals and therefore detect the heart rate. Tsuruoka et al. [25] presented in their paper a measurement of heart rate response using a portable pulse sensor system. They analysed the heart rate based on 1/frequency fluctuation and applying FFT spectral analysis.

#### 1.1.3. Translating Physiological Data to Signals, including Wearables

Early work on signals representing physio indicators focused on communicating the signal from the measurement device, such as a respiration belt, to screen-like devices.

The first wireless heart rate monitor was created in the early 1980s by Polar Electro. It consists of a chest strap transmitter and a wrist worn receiver [26,27]. This technique is still currently used to give athletes real time feedback about their breathing during their physical activities (walking or running).

A related system is deployed in running treadmills. While running on a treadmill, the user can monitor his heart rate, while looking at its screen in real time. There are two ways the treadmill can gather the information: the first, is using the handgrip, which is not very practical while running or walking, and the other one is connecting a wireless chest strap to the machine [28].

A different line of work is presented in Mauriello et al. and in Schneegass et al. [29,30]. Both of these studies demonstrated the communication of heart rate signals on shirts. The studies report an integration of the measurement sensor into a shirt together with a numerical display on the same shirt. [29] use a wireless chest worn monitor by Polar to measure the heart rate. Note that this integrated device reports the actual value and does not address deviation from an observed or learned normal behaviour. In Breeze [31], the authors discuss how presenting physiological signals as biofeedback to users raises awareness of both body and mind. The authors describe how breathing signals can be conveyed using a pendant-like device through three different modalities: visual, audio and haptics. They further evaluated how the breathing patterns were interpreted in a fixed environment and found that participants intentionally modified their own breathing to match the communicated biofeedback.

In [32], the authors discuss the use of auditory biofeedback in the context of stress management. According to the breathing rate, they adjust the quality of a music recording. If the breathing rate is optimal, the music is played with no additional noise. Otherwise, the quality of the audio decreases as the breathing rate gets further from the optimal breathing.

Another study [33] integrated respiration pacing methods into the computer desktop to influence to the respiration of the user. When the breathing rate is too fast, a horizontal grey bar appears on the screen and goes up (inhalation) and down (exhalation), representing the user’s target breathing rate. This allows the user to have a visual biofeedback.

Cibis et al. presented in their study [34] a ECG analysis system for scuba divers that would alert the diver in real-time. They placed ECG electrodes on subjects, as well as a smartphone placed in a waterproof bag, and determined for each person a resting heart rate. Whenever the heart rate of the diver was 40 bpm higher or lower than its resting heart rate, the smartphone would vibrate and create an alarm noise, alarming the diver that his heart rate is not normal.

Miri et al. presented PIV [35], a personalised breathing pacer that helps reducing anxiety using vibration patterns. They put in place two stressor periods, where participants had to answer some questions with a 9 second limit per question. On the second period, the PIV product was activated for the treatment group, and displays vibrations on the abdomen that represent inhalation and exhalation. They found that it does help reduce anxiety but it is hard for participants to adapt their breathing rate to the vibrations.

#### 1.1.4. Using Machine Learning to Analyse Physiological Data

The use of machine learning on physiological data allows us to train models which can be specific for every person and therefore get a deeper understanding and interpretation of signals in a personal context.

In [36], the authors use machine learning algorithms to detect whether a person is talking or not, using respiratory signals. They tested four different machine learning algorithms for their project: random forests, neural networks, support vector machines and linear discriminant analysis. After testing those algorithms, they noticed that the random forest gave them the best performance with an AUC value of 0.90. They had 15 participants in their study which lasted 1.5 h. The training set consists of the first 70% of the data of each participant and they used Gini importance to figure out the best features for their algorithm. They finally kept 13 features such as Ratio beyond sigma or Symmetry looking. They then split the data of each participant by activity and trained for each activity a model and evaluated using cross validation. Rezaei et al. [37] present a supervised machine learning algorithm to detect human gait phases. By gathering data recorded by an ankle brace, three different machine learning algorithms were tried: random forest, support vector machines and neural network. As in [36], they found out that random forest performs best and gave them an average accuracy of 95.49%.

Deep learning models on physiological data were also used for security related tasks. In [38,39], the authors present a breathing acoustics-based authentication for smart devices using Deep Learning models. They use breathing acoustics to allow user identification and user verification. The authors applied shallow classifiers (SVMs) and deep learning based classifiers (LSTMs). After analysis, they concluded that the LSTM models produce better f-scores with smaller storage required. Another user identification technique is using the ECG (electrocardiogram) signals. Lehmann et al. [2] present in their paper ECG to identify a person using machine learning methods. They had 20 participants that wore a chest belt tracking device for a week. They learned a random forest classifier, SVMs and a Neural Network. They found that random Forest Classifier performs better than the other two algorithms on the two scenarios that they tested. They concluded that ECG can indeed help identify a user but are less robust than expected.

#### 1.1.5. Summary

In summary, there are various technologies to support measurement of breathing rates and heart rates and much work has already been conducted in the context of using machine learning and statistical modelling on physiological data. In this project, we focus on analysing breathing rates and patterns to statistically model them and infer alert situations. We work on literature data as well as on data that we gathered for this study. The next sections present information about the respiration belt used for collecting our data and about how we analyse the outputs.

## 2. Materials and Methods

In this section, we describe the measurement methods and the data analysis steps we performed. The methods are described in a manner which is relevant to both the field study data and the literature data. Different parameters were used, as indicated.

### 2.1. Measuring the Respiration Rate: Respiration Belt

In this paper, we investigated the expansion of a respiration belt to capture the change in breathing rates. The respiration belt that we used, the “Alizé” breathing belt from the company “Ullo” [40], is a respiration belt that includes a rubber strip on its side. Whenever the person wearing the respiration belt inhales or exhales, the strip extends or relaxes and we capture those variations in real time. The respiration belt is placed below the sternum and over t-shirts (Figure 3). It is connected by a cable to our computer where we are collecting the data in real time. The data were captured with a sampling rate of 12 Hz.

### 2.2. Field Study: Using the Respiration Belt

In the following sections we describe how we collected data in our study. By gathering these data, we aim to answer several questions: what do the breathing rates/patterns look like while doing different activities for each participant? How can we analyse these data to identify outliers and generate alerts?

#### 2.2.1. Participants

Fourteen people participated in this study. The participants were between 20 to 74 years old, with an average of 36.79 and standard deviation of 16.04. Five participants were female and nine were male. Each participant was asked to complete a consent form and a questionnaire. All participants were in normal health condition.

#### 2.2.2. Procedure

The study protocol had a duration of 15 min per participant, including the setup time. The participants were asked to wear a respiration belt that was placed on the lower sternum and in a comfortable way so that they could still breath and act normally. The participants were asked to perform several activities for a certain amount of time. The experiment proceeded as follows: for 2 min each time, the participants were asked to sit, walk, run, and finally sit again. For every set of 2 min, they were allowed to talk only during the second minute. Figure 3 describes the design and an example of the 12 HZ sampled signal. In total, we collected for each participant around 12 × 60 × 8 = 5760 values.

#### 2.2.3. Datasets

We used the data as described above, collected for this study.

We also used data from the DEAP dataset [41], where they had 32 participants that wore a respiration belt for 63 seconds while analysing their ocular movement (among other physiological variables). The participants were watching small videos for the experiment. We will refer to these datasets as the field study data and the literature data, respectively.

### 2.3. Data Analysis

In this section, we describe all data analysis steps involved in learning distributions and generating alerts. These steps are also used to evaluate our methods as described in the result Section 3.

#### 2.3.1. Step 1: Saving Files

Our first step is to transfer the data from the respiration belt to our computer. We followed the script given by the manufacturer to connect the belt to the computer, and saved each participant’s data as a csv file. This step was only relevant for the field study.

#### 2.3.2. Step 2: Preparing the Training Data

We trained an alert generation algorithm for each person separately. In the field study, for each person, we had approximately 5760 values that represent 8 mins, where the data were down-sampled to 12 Hz. We trained only on the data where we knew that the user was not making any particular efforts, meaning the sitting phase and the walking phase. In the field study, we trained and tested the algorithm three times: first, we considered only the non-talking periods, then the talking periods, finally, we used a combination of the two datasets that we will call the combined data. In the literature data, we had 8064 values per person. We trained on the first half.

We used s(i), *i* = 0, …, 5759 and s(i), *i* = 0, …, 8063 to denote the raw measured signal for the field study and the literature data, respectively.

For our first analysis, we wanted to detect whether the breathing was regular, too low or too deep. We wanted to characterise, for each person, the basal distribution of their breathing signals. This would allow us to detect outliers and produce alerts when the breathing was too shallow or too deep.

#### 2.3.3. Step 3: Moving Maximum Algorithm, to Detect Deep Breath

When the breathing is too deep, the torso becomes larger and therefore the respiration belt extends more. To detect when this happened, we used the moving maximum of the breathing signals. For this part, we trained only on the sitting and walking stages of our study. We want to understand the regular breathing patterns with no activity recorded. Let us first start with the moving maximum. In the field study, we approximately trained on the 2880 first values, which represented the walking and sitting phases. We decomposed those values into windows of 12 values, which represented 1 s each. For each window, we kept only the maximum value M(i), yielding the moving max signal, as represented in Figure 4.

We then considered all the numbers M(i), *i* = 0, …, 2868, as obtained above, which allowed us to compute a distribution and then detect outliers. Outlier detection for the moving maximum allowed us to generate alerts designed to capture when breathing is too deep.

Let μ(M) and σ(M) be the mean and standard deviation of the above distribution. Note that these values pertain to a single subject in the cohort (a participant). A signal M(i) would be considered an outlier if, for example, M(i) < μ(M)−ασ(M) or M(i) > μ(M) + ασ(M) for some α > 0. We set the value α to represent the number of standard deviations we wanted to consider to define outliers. We set α to represent such a deviation for which we assumed greater variations to be personal. That is, all device variation was represented as part of the inferred distribution. The parameter α was also used to determine alerts for other parameters described later in the manuscript.

Whenever the detected signal M(i) > μ(M) + ασ(M), we identified it as a deep breathing and generated an alert. An example of a distribution with standard deviations is represented in Figure 5.

For the literature data, we used a similar procedure. Training was performed on the first 4032 values. Moving max was calculated for windows of size 128.

#### 2.3.4. Step 4: Moving Minimum Algorithm, to Detect Shallow Breath

We define the breathing as being too low when people are exhaling all their air, making the torso smaller than in regular breathing. We used the moving minimum to detect those irregularities. We did the exact same thing as presented above except we used the minimum values instead of the maximum. Low breathing was detected whenever the signal m(i) < μ(m)−ασ(m), for some α > 0, meaning that the size of the torso was exceptionally small. The results are discussed in Section 3.

#### 2.3.5. Step 5: Measuring Amplitude

We then calculated the amplitude of the breathing by subtracting for each person their moving minimum from their moving maximum. We thus got an amplitude value for each window. The window size was still 12 in the field study and 128 in the literature study. This is the main algorithm that we focused on. The amplitude of breathing was considered an outlier whenever the signal A(i) > μ(A) + ασ(A).

#### 2.3.6. Step 6: Fourier Transform, to Detect the Frequency of the Breathing

For our second analysis we used the Fourier transform. We trained on the sitting and walking phase signals, and tested on the running phase signals. As before, we trained three different times; one where the participants were talking, one where they were not, and one with the combined data. The Fourier transform represents the different frequencies that compose a given signal. By taking the absolute value of this transform, we have all the possible frequencies and we want to select the one with highest coefficient. We thus obtained the dominant frequency for every given window. Let us set the window size to 48, for example. We first define a window of 48 values starting at *i*:$$w_{i}=\left(s\left(i\right),s\left(i+1\right),...s\left(i+47\right)\right)$$
and then compute
$$f\left(i\right)=\underset{\left(0\leq j\leq 47\right)}{argmax}\left|{\left(FFT\left(w_{i}\right)\right)}_{j}\right|$$

We used window size = 48 for the field study data and window size = 128 for the literature data.

For each window, we then used this process to represent the most relevant frequency in that window.

We now consider all the numbers f(i), *i* = 0, …, 2831 or *i* = 0, …3903 for the field study and the literature data respectively, which allowed us to compute a distribution, and then detected outliers. Outlier detection allowed us to generate alerts when the breathing is too fast or too slow.

Let μ(f) and σ(f) be the mean and standard deviation of the above distribution. Note that these values pertain to a single subject in the cohort (a participant). A signal f(i) will be considered an outlier if, for example, f(i) < μ(f)−ασ or f(i) > μ(f) + ασ(f), for some α > 0.

In the example that we focus on, we considered only the case where the breathing was faster than regular, so we considered only when f(i) > μ(f) + ασ(f). An example is shown in Figure 6.

In the field study, we used a 5% threshold to define outliers.

#### 2.3.7. Step 7: Bivariate Distribution

In this part, we want to understand the relationship between the breathing rate and the ocular movement and generate alerts based on both. We only applied this analysis to the literature data, where they had 32 participants. We wanted to characterise, for each person, the basal distribution of their breathing and ocular signals. We can use this bivariate distribution to then generate alerts. For each person, we had 8064 values that represent 63 s, where the data were down-sampled to 128 Hz.

As in the previous section, we performed the moving maximum and the Fourier transform analysis on the data. In this case, we trained on the first half of the data and tested on the second half. We then constructed the bi-variate distribution, meaning that we wanted to detect whenever there were outliers both in the respiration rate and the ocular movement. As before, we used the learned distribution to derive alert thresholds, which we then tested on the second part of the signal.

For the training, we added jitter to our data points to make the signal less uniform, then we calculated the thresholds by defining the extreme 5% as outliers. We then placed the 5% thresholds on the testing data, as shown in Section 3.2. Whenever values were outside the frame, they were considered as outliers and would lead to alerts.

## 3. Results

We evaluated our methods by analyzing two datasets as described above. This section describes the results of our evaluation.

### 3.1. The Field Study

We run the algorithms on three different training/testing sets. We train and test on participants whenever they are not talking, train and test whenever they are only talking, and finally on the combined data. The training part always and only consists of the participants sitting and walking, and the testing part is when the participants are running. We train a model for each person separately and individually and infer distributions, on an individual basis, from personal data. In our case, the running episodes are used as simulated stress conditions. Running is considered an outlier situation when the distribution is inferred, personally per individual, from sitting and walking.

Recall that in this dataset we had a training signal of approximately 2880 values per sample. The testing signal is when the participants are running, which amounts to approximately 1440 values.

By mapping any new testing signal to the learned distributions we can determine whether the signal is an outlier or not. We note that this event occurs in several windows for the participant represented in Figure 7.

Upon applying the moving maximum threshold learned from the training on the walking/sitting phases to the test data collected in the running phase, we obtained the results presented in Figure 7. Figure 7 represents an example from one participant, when s/he is both running and not running and using the combined data of the talking stage and not talking stage. The percentage of alerts generated for this person, when running, using the sitting/walking signal as the basal distribution with signalling an outlier at α = +1.5 standard deviations away from the mean is 80.7% while talking, 95.1% while not talking and 85.1% when not separating the data by talking activity. The mean fraction of alerts generated for all participants when they are not talking was 0.355 with standard deviation of 0.320, when they are talking was 0.265 with a standard deviation of 0.332 and without separation was 0.288 with standard deviation of 0.310.

Upon applying the moving minimum threshold learned from the training on the walking/sitting phases to the test data collected in the running phase, we obtained the results presented in Figure 8. Figure 8 represents an example from one participant. The percentage of alerts generated for this person for α = +1.5 standard deviations away from the mean is 34.9% while talking, 84.7% while not talking and 48.8% for the combined data. The mean fraction of alerts generated for all participants when they are not talking was 0.123 with a standard deviation of 0.217; when they are talking was 0.087 with a standard deviation of 0.144 and when using the combined data it was 0.096 with standard deviation of 0.147.

Upon applying the amplitude threshold learned from the training on the walking/sitting phases to the test data collected in the running phase, we obtained the results presented in Figure 9.

Figure 9 represents an example from one participant. The percentage of alerts generated for this person for α = +1.5 standard deviations away from the mean is 51.1% while talking, 77.7% while not talking and 56.9% while using the combined data. The mean fraction of alerts generated for all participants when they are not talking was 0.224 with standard deviation of 0.213, when they are talking was 0.183 with a standard deviation of 0.230 and when using the combined data it was 0.187 with standard deviation of 0.204.

We present in Figure 10 the fraction of alerts that we get for each participant for the amplitude calculations. For 12 out of the 14 participants, we observe that the amplitude alerts are mostly to the right of the mean, except for cases 0 and 11. Indeed, we expect the amplitude alerts, when running, to be on the right of the mean.

### 3.2. The DEAP Dataset (The Literature Data)

Recall that for the DEAP dataset, we use two physiological signals: the respiration rate and the ocular movement. Both signals have 8064 values. We use the first half of the values for training, and the second half for testing. After learning thresholds from the training data, we create a histogram to see where the testing data are compared to those thresholds. We can see an example of the graphs including the standard deviations for the respiration rate in Figure 11. Note that in the DEAP data, there is no difference in the characteristics between test and training, in contrast to the field study case (where training was sitting and walking and test was running).

To get the distributions of frequencies of the signals for the respiration rate and the ocular movement, we use the Fourier transform. Figure 12 represents the results of the Fourier transform for the respiration rate.

After performing the Fourier transform and getting results for both the respiration rate and the ocular movement, we trained a bi-variate distribution as shown on Figure 13. We defined thresholds so that 5% of the training values are outliers. By putting our thresholds on our testing set of the two variables, as shown on Figure 14, we obtain a total number of 95.1844% of the data points that are inside the threshold, and the rest are outside. It makes sense since in DEAP, training and test are of the same type.

## 4. Discussion

### 4.1. How to Use the Algorithm to Generate Alerts to Be Displayed on a Wearable Device

Recall that the methods presented here are designed with the potential application, in mind, of actually displaying the alerts on a wearable device. To actually use the algorithm in this manner we need to train and learn distributions for potential users. First, a respiration belt is set up on the prospective user and captures their breathing for a certain amount of time. Once we have these data, we analyse the regular breathing rate and obtain the statistical model for that. Similar to the data presented for example in Figure 7 and Figure 11.

Once the training is finished, the algorithm is ready to be used. The user wears the garment that is connected to the respiration belt. The respiration belt is sending in real time the data it measures to the microcontroller which analyses the breathing rate. According to the model obtained in the training part, the algorithm is able to analyse whether the breathing isn’t normal. Or, in other words, to detect outliers. While the breathing is normal, the microcontroller makes the LEDs blink regularly, which, in turn, makes the fibre optics blink regularly. If the breathing is too fast, the LEDs blink faster, or change colour, so the fibre optics blink faster, or change colour. In this case, both the person wearing the garment and the observers notice the outlier behaviour. An early prototype is shown in Figure 15 and Figure 16.

### 4.2. Considerations

In some of the methods we described above, we need to determine a window size with which to work. For example, in the field study, when working with the amplitude, we had set the window size to 12. This is done for two reasons. First, 12 samples correspond to an actual time of one second. In addition, we note that counting amplitude alerts to the right of the mean (RA) versus amplitude alerts to the left of the mean (LA), in running, we get the following number of RA > LA events. When using a window of size 2 or 3, we get 11/14. When using a window size between 4 and 19, we get 12/14. When using a window size of 20, we get 11/14 and when using a window size of 40, we get 8/14. Therefore, our selected window size of 12 also corresponds to the middle of the preferred performance. Other parameters such as α can also affect the performance, especially in terms of specificity and sensitivity.

### 4.3. Limitations

An important caveat about this study is that the data are still small in scale and that more field studies need to be conducted in order to improve performance and to evaluate practical aspects. In particular, we note that the variation in the percent alerts generated, as reported in Section 3.1, is fairly high.

As indicated above-training and learning need to be executed on each individual separately. We can not expect learning from a population to be applicable to other individuals. This limitation may be obstructive, but not a show stopper, in practical terms. Moreover, we expect that training and modelling can be different for different devices, in particular-different respiration belt manufacturers. It is therefore important to train for the specific configuration that is expected for the actual usage scenario.

Finally, we emphasise that this work is only at the level of a proof of concept and that the study was conducted on a small cohort.

### 4.4. Applications

Garments, or wearable devices, that can display alerts as generated by the system we described herein, can have a variety of applications - health monitoring, hazardous work environments, space exploration, scuba diving and others. The physiological information gathered by the sensors of the wearable system are translated through the smart fibre optic textiles and presented as a visual display on the garment and can be seen by observers. From there, the machine learning embedded into the translation creates the signal of regular or irregular breathing. This is then interpreted by the observers to drive necessary actions such as assistance to the device user.

Children below speaking age can not provide verbal alerts on physiological situations. Even in older ages its is often the case that children, as well as adults, may be deeply engaged in some activity, in such a way that distracts them and thereby also renders verbal alerts not possible. In such cases it can be useful to have a system that automatically generates alerts, as we describe in this work.

Another specific example is scuba diving. Clearly, the divers are potentially exposed to physiologically extreme conditions and are not necessarily able to report on their condition. Our proposed system can be useful in this application, as well.

## 5. Conclusions

We presented methodology and results for detecting episodes of extremely fast or extremely slow breathing using a machine learning approach. We used data from a respiration belt for our study. Potentially, this method can be used as part of a system that includes a wearable device. The generated alerts could be sent to a wearable device which would display different patterns according to the breathing rate/pattern of the user and according to the alerts generated. In this project, we develop a methodology for monitoring the physiological status of a user. In future work, we will extend the project to suggest design patterns and adequate materials for visualising the alerts.

Our next step is to complete our implementation and then to test our prototype and see how people will react to it. We also plan on expanding to more physiological data, such as the heart rate, the temperature of the body or the oxygen level in blood and develop a distribution modelling approach for each physiological data type. We will learn also multidimensional distributions representing the basal behaviour for any single individual and infer outliers under this multivariate model. 

## Figures and Tables

**Figure 1 sensors-22-06306-f001:**
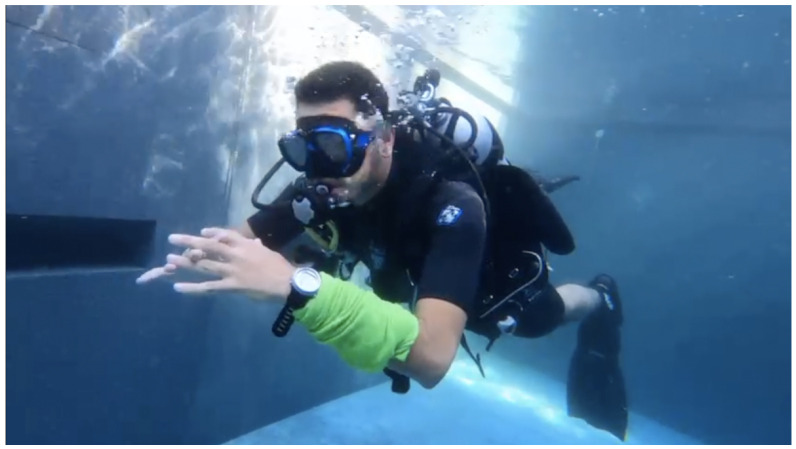
Visual representation of what we envision the system to look like during regular breathing.

**Figure 2 sensors-22-06306-f002:**
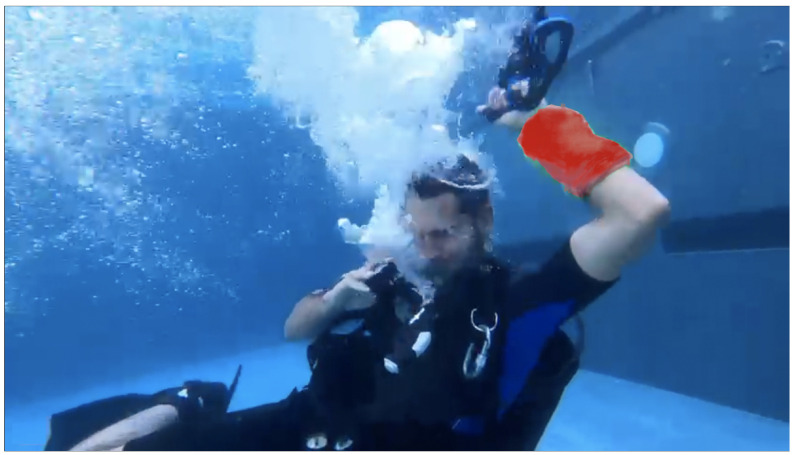
Visual representation of what we envision the system to look like when an alert is generated and transmitted.

**Figure 3 sensors-22-06306-f003:**
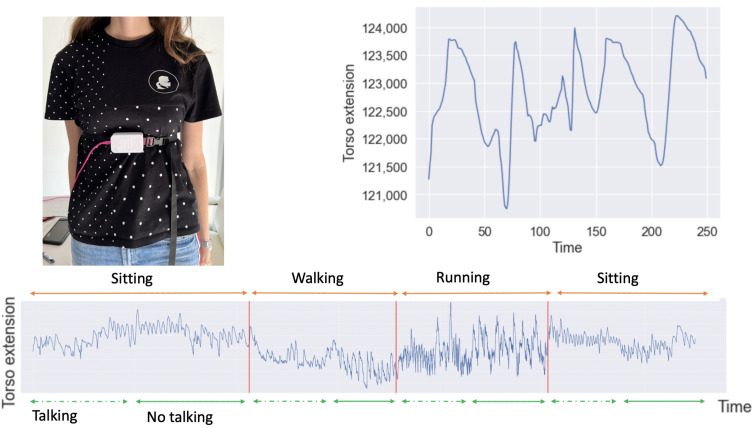
Participant wearing the “Alizé” respiration belt and the torso extension signals obtained during the experiment.

**Figure 4 sensors-22-06306-f004:**
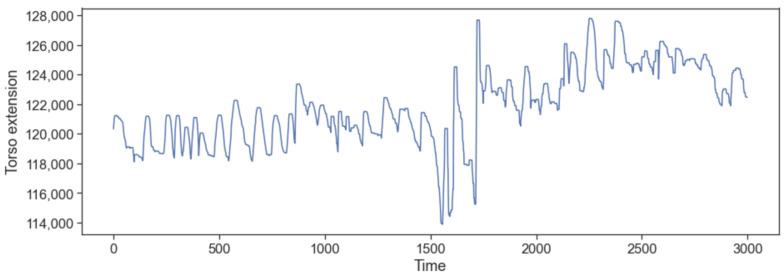
The moving maximum signal as generated from one participant, in the field study data, during the walking and sitting periods, including the talking and non-talking segments. The window size used for calculating M(i) is 12 in this case. Every time unit corresponds to approximately 0.08 s.

**Figure 5 sensors-22-06306-f005:**
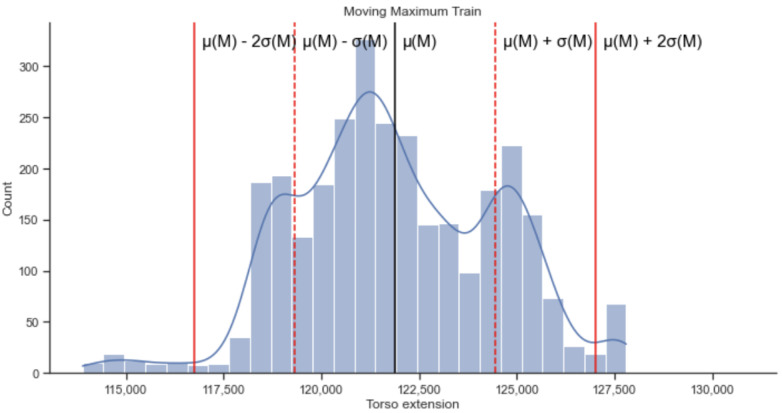
A histogram of moving maximum values for the training data. The black line represents the mean of the distribution, the dashed red line represent 1 std away from the mean, and the thick red line represent 2 std away from the mean.

**Figure 6 sensors-22-06306-f006:**
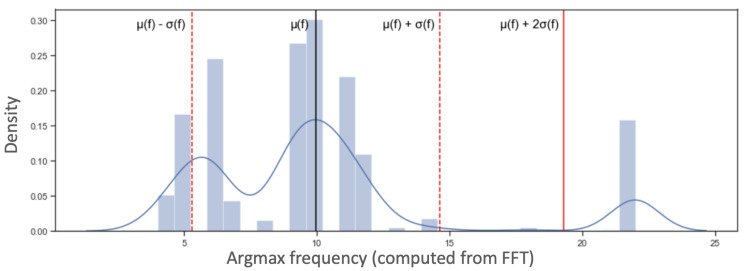
A histogram of values of the representative frequency (computed from FFT, see text) for windows of size 128, in the training data of the respiration belt, DEAP dataset.

**Figure 7 sensors-22-06306-f007:**
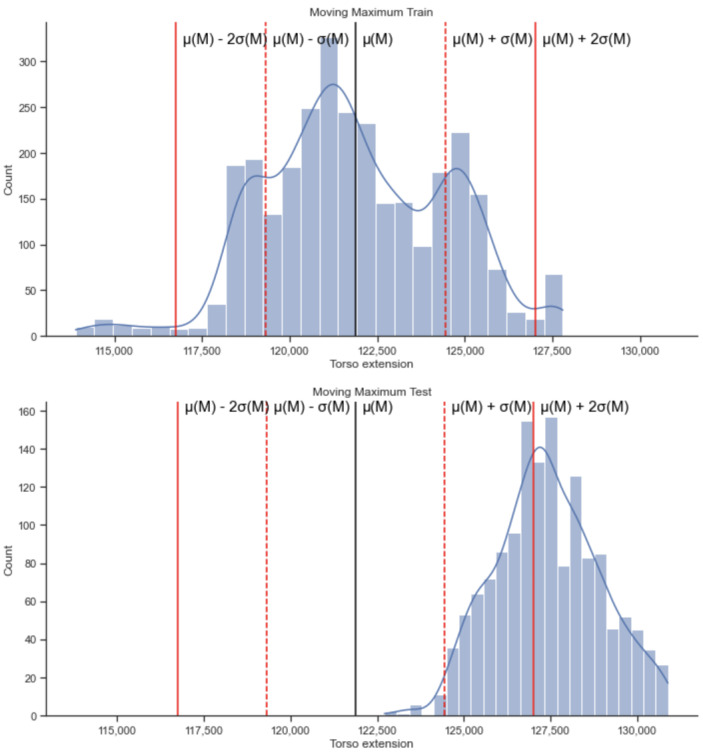
Distribution of training (sitting, walking) and testing values (running) for the moving maximum algorithm on the respiration signals for a participant on the combined data (talking and not talking).

**Figure 8 sensors-22-06306-f008:**
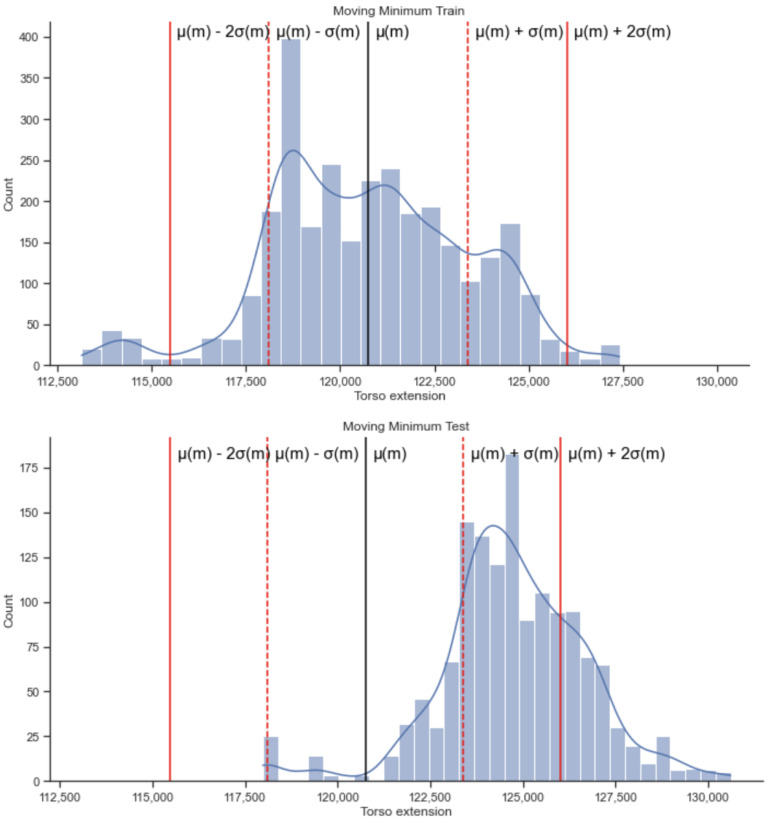
Distribution of training (sitting, walking) and testing values (running) for the moving minimum algorithm on the respiration signals for a participant when using the combined data (talking and not talking).

**Figure 9 sensors-22-06306-f009:**
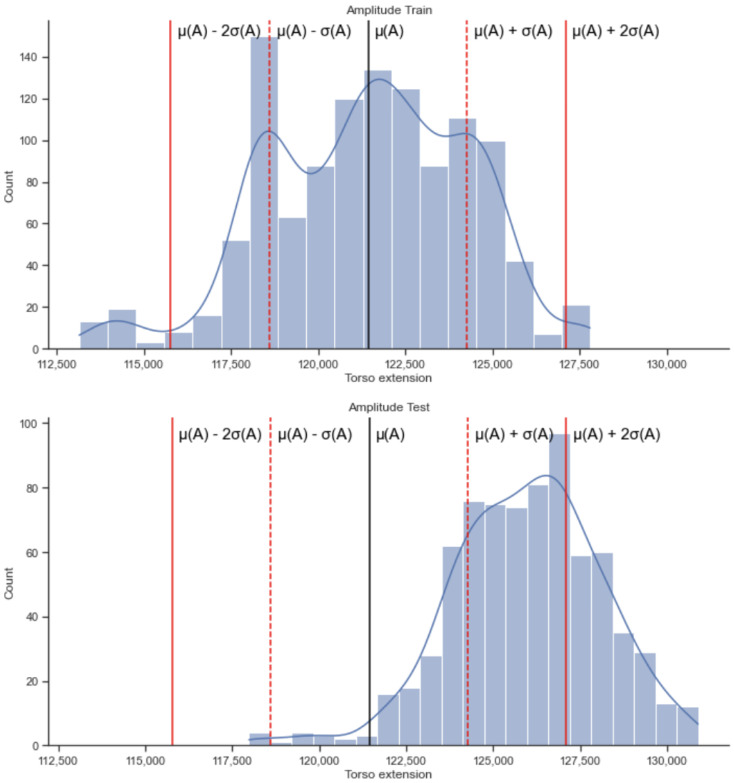
Distribution of training (sitting, walking) and testing values (running) for the amplitude calculations on the respiration signals for a participant when using the combined data (talking and not talking).

**Figure 10 sensors-22-06306-f010:**
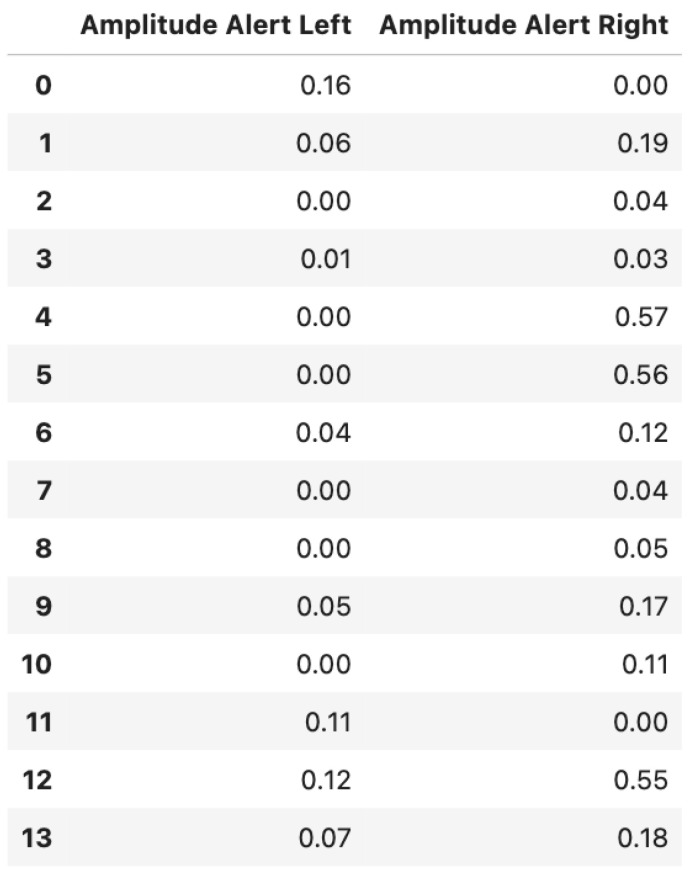
Alerts (fractions) for the amplitude for each participant for the combined data.

**Figure 11 sensors-22-06306-f011:**
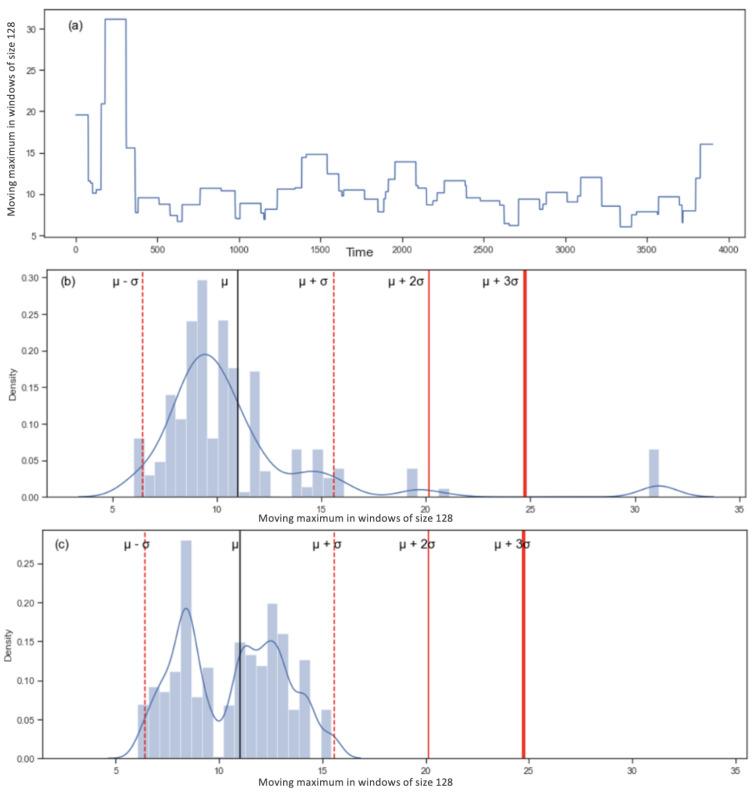
The upper panel (**a**) represents an example of the moving maximum for each window of 128 values for the breathing signal. The middle panel (**b**) represents the training distribution of the values in (**a**). The values between the two red lines μ−σ and μ + σ (around 5 and 15) represent normal breathing. The values that are on the left of μ−σ (0 to around 5) represent shallow breathing and the values that are located at μ + σ and more represent deeper breathing. The lower panel (**c**) represents the testing distribution compared to the threshold defined in panel (**b**).

**Figure 12 sensors-22-06306-f012:**
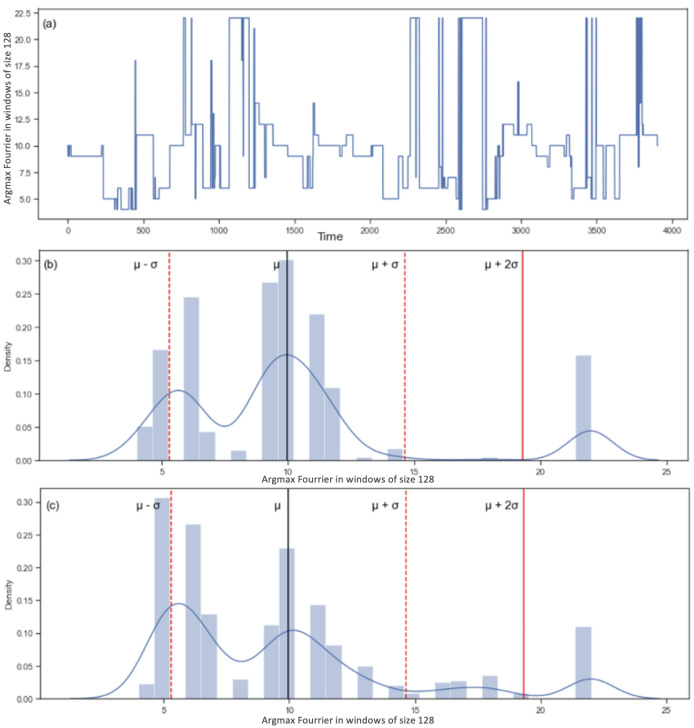
The upper panel (**a**) represents the argmax of the absolute value of the Fourier transform for each window of 128 values. The middle panel (**b**) represents the training distribution of the values in (**a**). The values that are between the two red lines μ−σ and μ + σ (around 5 and 15) represent normal breathing rates. The values that are on the left of μ−σ (0 to around 5) represent slow breathing rate and the values that are located at μ + σ and more represent fast breathing rate. The lower panel (**c**) represents the testing distribution compared to the threshold defined in panel (**b**).

**Figure 13 sensors-22-06306-f013:**
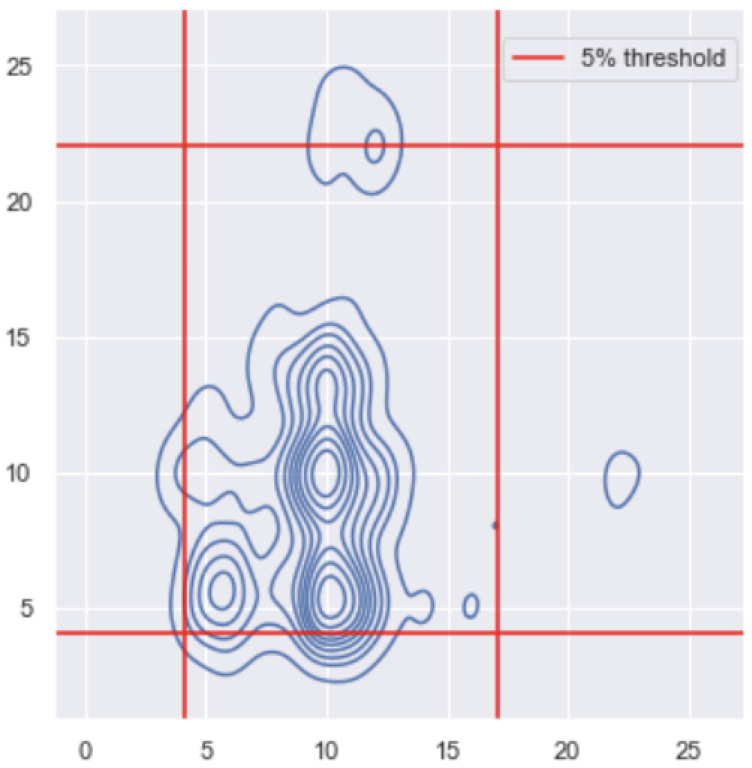
Distribution of the training bivariate Fourier values of both the respiration and the ocular movement and the threshold of 5% (red lines) of the values to be defined as outliers.

**Figure 14 sensors-22-06306-f014:**
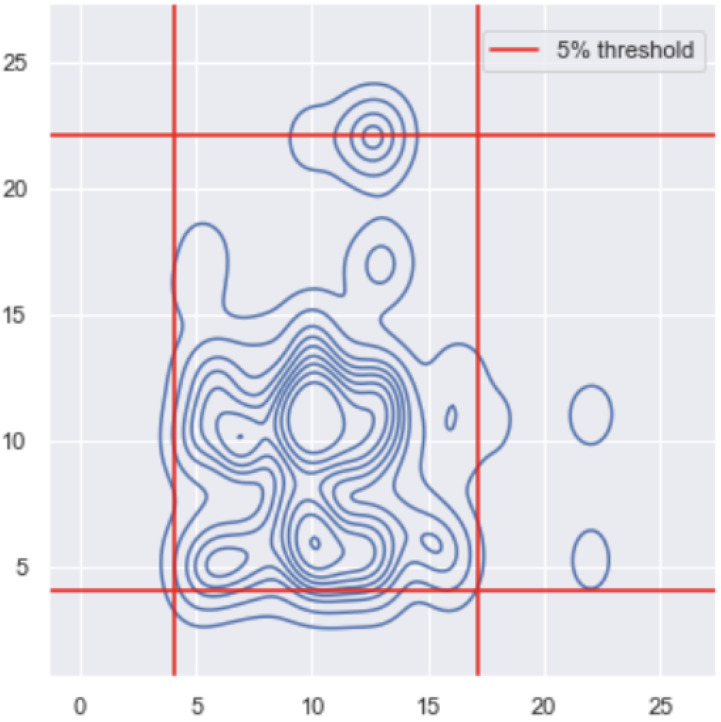
Distribution of the testing bivariate Fourier values of both the respiration and the ocular movement and the thresholds (red lines) as defined as Figure 13.

**Figure 15 sensors-22-06306-f015:**
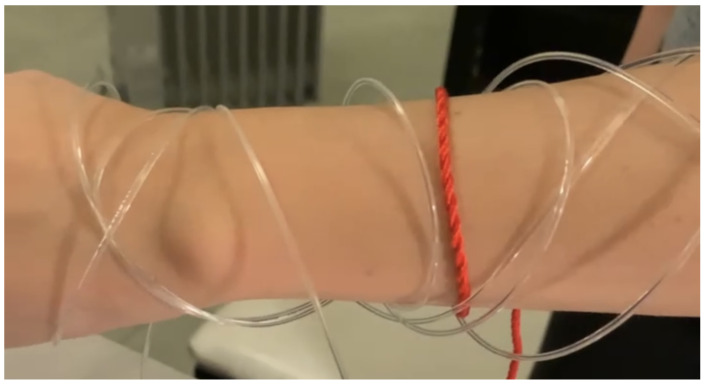
Early prototype, normal condition.

**Figure 16 sensors-22-06306-f016:**
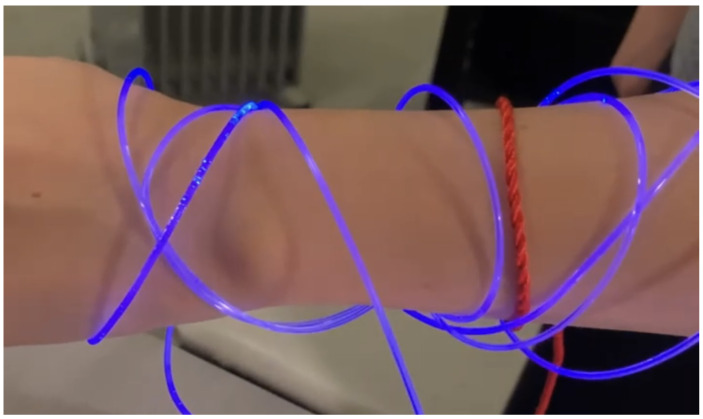
Early prototype, an outlier.

## Data Availability

Data and code available upon request.

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
