# Peer review of "Generating Alerts from Breathing Pattern Outliers"

_sensors, 2022, doi:10.3390/s22166306_

Round 1

Reviewer 1 Report

The paper presents the methodology for monitoring the physiological status of a user based on breathing pattern outliers. The paper is well-written and well-organized.

 The aim of the paper and the contribution (novelty) should be separately described.

The literature review is done, but more papers presenting results of breathing data (and maybe other biomedical signals as well) analysis – including statistical analysis should be described. The novelty of the paper should be clearly defined in the context of the existing state of knowledge.

 Fig 1 and 2 – is this prototype of authors or photos taken from outside? The prototype presented in Fig. 15-16 looks different.

2.2.2 – How the data were trained?

Did you consider any breathing norms dedicated for a particular user profile (for example according to the age?

How do authors want to distinguish outliers coming from equipment from those coming from the user?

What was the “ocular movement” exactly and how they were analyzed?

Units and axes descriptions should be added to figures

Author Response

We highlighted all text added to this new version. We also modified the abstract. 

Reviewer 2 Report

The study proposes an algorithm for detecting outliers of breathing rate. The manuscript is clearly written but there are major comments.

The most important drawbacks are:

1.      The manuscript lacks an explanation of how the parameter α is estimated (Ln 288)

2.      Ln 276-277 states that the window size is 48 for the field study data and the window size is 128 for the literature data. What window size should we use if we want to test the algorithm on our data?

3.      Ln 312-313: it is necessary to additionally explain why in the training data are included only participants who sit and walk, and in the test participants who run.

4.      It is necessary to strengthen the Results section by comparing the results obtained by applying state-of-art methods from the literature on the observed data sets.

5.      Keywords should be selected appropriately.

6.      Ln 44-48 future work should be part of the conclusion, avoid this in the introduction.

7.      In the abstract, it is necessary to state the results obtained by the proposed algorithm clearly.

8.      Figures 3 and 4 are missing measurement units on the y-axis.

Author Response

(The authors gave the same response as above.)

Reviewer 3 Report

The paper presents a nice study, well written, and it fits well on journal scope. My only question is how precise is your technology against other methods on market?

Congratulations

Author Response

(The authors gave the same response as above.)

Round 2

Reviewer 1 Report

Thank you for resubmission, the paper is better now.

Author Response

Thank you for your review. 

Reviewer 2 Report

The authors did their best to address my revision comments. I find them entirely satisfactory, but I still think that study about the influence of window size on final results should be done. It should be part of this study, not future work. 

Author Response

Thank you for your review. We added an explanation of the considerations that went into selecting the window size. This is now a subsection in the discussion (highlighted in yellow). As indicated before, more work will be needed in order to even more carefully define the optimum parameters.